# Dairy Cattle Rumen Bolus Developments with Special Regard to the Applicable Artificial Intelligence (AI) Methods

**DOI:** 10.3390/s22186812

**Published:** 2022-09-08

**Authors:** Éva Hajnal, Levente Kovács, Gergely Vakulya

**Affiliations:** 1Alba Regia Technical Faculty, Óbuda University, 1034 Budapest, Hungary; 2Institute of Animal Sciences, Hungarian University of Agricultural and Life Sciences, 2100 Gödöllő, Hungary

**Keywords:** rumen bolus, data processing, artificial intelligence, machine learning, dairy cattle

## Abstract

It is a well-known worldwide trend to increase the number of animals on dairy farms and to reduce human labor costs. At the same time, there is a growing need to ensure economical animal husbandry and animal welfare. One way to resolve the two conflicting demands is to continuously monitor the animals. In this article, rumen bolus sensor techniques are reviewed, as they can provide lifelong monitoring due to their implementation. The applied sensory modalities are reviewed also using data transmission and data-processing techniques. During the processing of the literature, we have given priority to artificial intelligence methods, the application of which can represent a significant development in this field. Recommendations are also given regarding the applicable hardware and data analysis technologies. Data processing is executed on at least four levels from measurement to integrated analysis. We concluded that significant results can be achieved in this field only if the modern tools of computer science and intelligent data analysis are used at all levels.

## 1. Introduction

The worldwide trend is to increase the number of animals on cattle farms from an economic point of view, but animal welfare, sustainable farming, and the carbon footprint, to which animal husbandry is often linked in a socially controversial way, are also important key terms in our modern era. The welfare of animals can be ensured with the reduction in human labor input only with continuous monitoring with the appropriate sensor technology and the application of the related data processing, data integration, and intervention system. Recognizing this, the number of sensor types used in animal husbandry and the number of products implemented are constantly increasing. The used sensors show great variety.

The sensors can be grouped from several aspects, firstly according to the placement method. According to Knight’s classification [1], there are “at cow”, “near cow”, and “from cow” sensor systems. The first group (Figure 1.) includes sensors placed on the cow and in the cow (neck, legs, body, vagina, tail, rumen, rectum, and subcutaneous tissue); the second group monitors the interaction between the cow and its environment (external camera systems, drone technology, and weight measurement systems); finally, the third group includes systems for monitoring the physical and chemical properties of products of animal origin.

An important additional grouping aspect for “at cow” sensors (see Table 1) is the type of intervention required for placement, according to which sensors can be used without the animals being hurt or that require veterinary operation. The third aspect is the working interval of the sensor, which is due to the stability of the installation and the technical lifespan of the sensor. The focus of this review is on the first group of sensors, including the rumen boluses, highlighting data-processing issues.

In the case of dairy cattle, the application of sensor technology is a rapidly evolving field. A study published in 2010 on trends in dairy farming technologies does not even mention cattle-monitoring IoT techniques [2]. Similar studies in 2019 [3] and 2020 [4] clearly put their vote in favor of continuous monitoring of cows as one of the most important technologies proposed in cattle farming.

Animal-based sensors can be divided into two groups; there are devices for research purposes and commercially available on-farm decision support systems. Separation is not necessarily differentiated according to the level of development (according to TRL levels), but the purpose of the two sensor groups is different, and this is particularly pronounced depending on the IoT technologies used. One helps research tasks with objective measurements, and only a small proportion of them are currently dedicated to the day-to-day practice of dairy farms. Research-aimed sensors provide primary measured quantities, which will be evaluated and interpreted by the researchers according to the aspects of the actual research question and with statistical methodology [5,6,7,8,9,10,11,12].

At the same time, there are more expectations in dairy farms today. Tools and methods need to be developed to provide a complex expert support by assessing behavioral and physiological characteristics in a complex way, thus helping farmers to make decisions [1,4,13,14,15]. The devices must be usable throughout the whole life of the animals. Despite the importance of complex sensory monitoring, very few publications have reviewed these issues, which is why it was chosen as a topic, highlighting the specific issues of lifelong-monitoring data processing.

Historically, the first rumen bolus sensor measurements appeared after the 2000s in experiments. The first measurements were taken on reticulo-ruminal pH and temperature [9,14,16]. The experiments were typically performed on fistulated cattle and lasted for a few days. During this period, transmission and/or storage techniques at the given technical level have been developed. These typically did not allow long-term measurements. At this time, the study on the mechanical design of rumen boluses also began, and two types were distinguished: the winged and the weighted mechanic types [3]. Both methods served for stabilizing the position of the sensor, as unlike traditional sensor devices, these sensors are not fixed, which makes it more difficult to process and interpret data.

However, these early time sensors (bolus sensors as well as other “at cow” sensors) provided only primary information (lying time, average activity, or time spent ruminating, or ECG). It was not even possible to carry out long-term measurements with the technology; however, in 2010 [9], it was reported that no theoretical obstacle for lifelong monitoring exists using bolus methodology.

The current trend is sensor fusion or sensor integration [17,18,19]. This means that measurements are made simultaneously with several sensors, and these measurements are aggregated by a data-processing system from which not only integral data but integral statements regarding important issues related to the condition or welfare of the animals are gained. Such claims are the estrus, inadequate feeding, symptoms of disease or lameness, and prediction of the time of calving.

The aim of this paper is to give an overview of cattle rumen bolus-sensing technologies, and in the following chapters, it will discuss: (i) the areas of application of rumen boluses, (ii) the sensing solutions, (iii) data transfer solutions, (iv) data-processing and artificial intelligence (AI) solutions which are already applied or suggested in rumen boluses.

## 2. Areas of Application of Rumen Boluses

The use of rumen boluses in cattle farming is not widespread today due to the high cost of the sensors compared to external ones used in other smart herd diagnostic systems. In practice, they may have diagnostic value during the transition period in smaller farms to detect the ruminal adaptation to changes in close-up and lactation diets, as a number of production-related diseases may affect the pH within the rumen. The measurement of ionic activity has been employed to monitor the rumen status of cattle, particularly to detect sub-acute rumen acidosis (SARA). Several pH thresholds for SARA are reported, influenced by factors such as sampling location, sampling time, and measurement frequency. Considering its diurnal patterns [20], thresholds of a daily ruminal pH average <6.16 [21] and of a period with ruminal pH < 5.6 for more than 3 h/day [22] or ruminal pH < 5.8 for more than 5 h/day [21] were described for SARA risk cows using continuous measurements of the pH in the ventral rumen.

Several wireless intraruminal sensors were developed in the last decade, and due to the high correlation between sensory pH data and those obtained with calibrated laboratory pH probes, ruminal boluses became accepted tools for ruminal-pH measurement [23]. However, due to their extremely short lifespan for pH measurement, rumen boluses are currently used primarily for research purposes. These measurement systems are noninvasive alternatives to surgery using a pH probe calibrated with a reference solution, a processing unit that records the pH signal, and a converter that converts the pH signal to radiofrequency and a receiver [24]. Monitoring systems developed for working farms generally measure the pH in the reticulum. Most of the publications presented the reticulo-ruminal pH [25,26], while in other papers, authors distinguish between the ruminal and the reticular pH values [27]. Since the ruminal and reticular pH values have been proven to be different [28], the sensor localization was considered when establishing the diagnostic limit for SARA. The pH is higher in the reticulum; therefore, instead of ruminal pH < 5.8, a reticular pH of <6.04 was suggested for no more than 6 h/da [29].

In a recent study, a refined ruminal pH indicator was recommended in long-term SARA diagnosis based on individual reticulo-ruminal pH patterns, because commonly used pH SARA indicators were not able to distinguish SARA due to the great interindividual variability in ruminal pH, the calibration drift of the pH probe (0.025 pH units/week), and certain high-frequency noises causing false-negative pH peaks [30]. By correcting the absolute pH value with these factors, SARA detection could be improved significantly. Normalized pH kinetics were smoothed by applying a 180 min moving average, resulting in filtered normalized kinetics. Using the normalized pH indicators, it was more accurate to model the effects of feeding high- vs. low-starch diets as normalized pH parameters increased significantly [30]. In a more recent study, significant associations between reticular pH and chewing behavior (feed intake time, the frequency and time spent ruminating, and number of rumination boluses) and milk chemical composition (fat and fat/protein ratio) were found [31]. Still, the continuous on-farm reticular pH measurement has serious limitations, due to the limited lifespan of the pH boluses and costs involved.

Besides the assessment of ruminal pH, bolus sensors are also able to measure ruminal-temperature changes, reflecting a shift in animal physiological states [32,33,34,35]. It was shown by several authors that a decrease in ruminal temperature reflects drinking and feed-intake events, and its increase coincides with increased body temperature [33,36,37]. Monitoring changes in the ruminal temperature and the cows’ activity can facilitate the early detection of estrous [38] and inflammatory conditions [39], and is, therefore, a valuable approach to support herd management. Rumen temperature measured by bolus sensors was compared with rectal temperature measured using digital thermometers, and the PCC value from five independent study samples showed that the rumen temperature measured by the bolus sensors was moderately correlated with rectal temperature [40,41,42]. Similarly to the pH, ruminal or reticular temperature has been investigated as a tool for the remote measurement of core temperature [41]. Reticulo-ruminal temperature correlated with rectal temperature and respiratory rate in beef cattle [34] and has been investigated as an indicator of heat stress [35,43] and as a predictor of calving in dairy cows [26]. As reticulo-ruminal temperature is significantly affected by water intake [34,40], reticulo-ruminal boluses are able to monitor drinking events and investigate factors affecting drinking behavior in cattle. However, ambient temperature and cow identity should be also considered, as either cow threshold characteristics or ambient temperature affected significantly drinking events [44]. Moreover, feed intake and milking increase the frequency of drinking events in dairy cattle [45] and a reduction in water intake was reported to decrease milk yield by up to 26% [46].

## 3. Sensors Used in Boluses and for Related Experiments

The rumen bolus is also suitable for lifelong monitoring due to its stable placement and the long-life and small-sized batteries available today. In such a case, it is reasonable to measure as many physiological characteristics as possible with sensor fusion, which raises further questions. A rumen bolus is a heavily constrained device in terms of the available battery power, space, and bandwidth. It is usually watertight, which makes chemical sensors difficult to use. Finally, the price is also a limiting factor. On the other hand, during preliminary or short-time experiments, virtually any possible kind of sensors can be used. Here, the otherwise very important energy consumption, size, and price have secondary interest.

Rumen boluses are usually targeted to measure a subset of three main modalities; the ambient temperature, the motion activity (including rumination), and the pH. Temperature is one of the first non-electrical physical quantities, which could be transformed into an electrical signal and measured digitally. The most straightforward method utilizes the temperature dependence of different properties of PN junctions of semiconductors, making it possible to integrate the whole digital sensor into a single chip. Single-chip temperature sensors are available from numerous vendors in a wide range of accuracies for a moderate price. Other sensor types use the thermoelectric effect of a thermocouple or resistive thermal sensors, both combined with an analog–digital converter. These solutions provide a much wider thermal scale (way above 100 °C), which does not exist inside the rumen. Examples for using ambient temperature sensors can be seen in [32,47].

Measuring any factor related to motion is carried out by a MEMS (micro-electromechanical system) [48], or possibly gyroscopes. These sensors are often integrated in a single device, called an inertial measurement unit. These devices contain MEMS sensors, analog–digital converters, and the supporting logic to provide a standard digital interface. There are only a few companies providing cost-effective MEMS accelerometers or inertial measurement units with high sensitivity, selectable sample rate, and low power consumption. These versatile sensors can serve as a basis for several data-processing algorithms. Papers [7,12,48,49,50] show good examples for using accelerometers in reticulo-ruminal boluses.

Rumen pH is still very challenging to measure. Sensors appropriate for pH measurement use electrodes in direct contact with the rumen fluid; therefore, they are prone to corrosion, which dramatically shortens their lifetime to a few months. Experimental solutions are mentioned in [51,52].

During preliminary or short-time experiments, the goals, conditions, and available measurement methods are different from that given for a bolus. As measuring motion can provide meaningful information not only inside the rumen but in the body surface of the animal, accelerometers and IMUs are often used in preliminary experiments as well, a typical application being a neck collar or fitted to the animal’s leg. Accelerometer-based data loggers attached to the leg can provide information about the lying behavior of the animals [53].

Finally, we can mention the special sensors, which are primarily used in medical diagnostics, such as ECGs [8,48], pulse oximeters, and respiration transducers.

## 4. Data Transfer Solutions

Transferring the sensed raw or preprocessed data has key importance on the system performance. Each kind of data transfer method has different limitations and gives constraints in terms of communication speed, power consumption, range, cost, and complexity, just to name a few. Existing systems and experiments may use two different approaches. Data can either be recorded inside the bolus or forwarded to some kind of receiver using a wireless channel. The former method can be used universally, while the latter one can be applied only in the case of fistulated cattle.

The advantage of storing recorded data inside the bolus is the high available capacity, which allows continuous measurement with high sample rates even with multiple sensors. The most ubiquitous storage media are different SD cards [53]; however, the storage medium was not mentioned explicitly.

Regarding the application of an intra-ruminal telemetric sensing device, there may occur some discontinuity in data transmission caused by the distance between the animal and the receiver or the increased signal degradation due to the lying position [32]. In adult cows, an increased packet loss for weighted ruminal boluses was also found [54].

Year by year, new radio communication solutions are developed and this progress can be also observable in bolus research. The first experiments date back to 1974, when Hanton and Leach [55] developed the first RFID system. In the field of dairy cattle farming, automatic milking systems were the first adapters of this new technique [56], followed by other precision systems, where RFID was used to identify each animal for the robotic arms in [49], where ear tags were used to monitor the feeding behavior of the animals. The main advantages of passive RFID tags are the low price and the low energy consumption. Sensing is only possible with active RFID technology [52], but its communication range is very limited [55].

For digital radio communication, two possible directions are available. Either an existing standard can be used, or a different solution can be developed based on the existing radio chips. The two most well-known radio communication standards are 802.11 (WiFi) and 802.15.1 (Bluetooth). Both use the 2.4 GHz industrial, scientific, and medical ISM band; therefore, they are prone to the absorbing effect of water. These solutions are easy to interface to a laptop or a handheld computer [55]; however, the communication is limited to the vicinity of the examined animal. Another important drawback of using WiFi or any Bluetooth standard besides bluetooth low energy (BLE) is the relatively high power consumption.

There are several different digital radio communication modules operating in various frequencies with different modulations. Transceivers operating in the UHF (ultra-high-frequency) ISM bands (315, 434, and 818 MHz) require larger antennas than their 2.4 GHz counterparts, but the absorption is much higher in the latter case.

Mottram et al. [9] used the 433.92 MHz frequency band with a moderate 4 mW power to provide a bidirectional data link between the bolus sensor and the base station. The antenna of the bolus sensor was an inefficient helical one; the handheld unit was equipped with a quarter-wave whip antenna.

In Ref. [57], a special workaround was proposed to compensate for high attenuation of the body of the animals. They used neck collar transponders to extend the communication range of the bolus sensors. In Ref. [32], a similar method was described, where the 433 MHz band was used and an additional transponder was placed to one of the legs of the animals.

Many of the aforementioned solutions used one of the built-in wireless communication protocols of the computers of that time. Although the application of those solutions seems to be obvious, their parameters are not optimal to use in a bolus. The bandwidth of WiFi and even Bluetooth is unnecessarily high, while they are quite limited in terms of communication range or the number of simultaneously handled devices. The other generally available ad hoc solutions had more or less issues as well.

In the last decade, several wireless standards have been developed, many of them specially tailored to serve the requirements of IoT applications and LPWANs (low-power wide-area networks) [58]. The most widely used and well-known protocol is LoRa [59]. The first LoRa standard was issued in 2015 [60] and became widespread in the next few years. LoRa targets applications which require very low data rates (even a few bytes a day, down to 146.1 bps) and have limited energy. The most remarkable property of this protocol is the great link budget (up to 150 dB), which allows long-range communication and challenging propagation circumstances with high attenuation. LoRa offers modulation settings with different data rates to balance between power consumption and link budget. LoRa can operate on 169, 433, 868, and 915 MHz, based on the local regulations.

Although LoRa can be used alone with local receivers (gateways) for experimentation and gives independence from service providers, LoRaWAN gives much convenience through well-designed upper network layers and optional cloud support. Different commercial LoRaWAN networks are operated worldwide in several countries and provide good coverage.

SigFox [58] was issued a few years later than LoRa and offers a promising commercial communication solution for low-power, low-data-rate applications. It uses a 100 or 600 bps data rate and provides a link budget even better than LoRa (around 160 dB). SigFox operates from 862 to 928 MHz.

NB-IoT [61] is operated by the cellular providers as a part of the LTE (4G) network. It operates from 700 to 2100 MHz and authorizes the device with a SIM card, similarly to any cellphone. NB-IoT has the highest maximum link budget (164 dB) out of the last three technologies and provides the highest bandwidth (26 kbps).

The future of bolus sensors is obviously in using one of these IoT communication protocols.

## 5. Data Processing and Artificial Intelligence Methods

The data-processing methods of experimental rumen boluses used for research tasks is the application of statistical methods used widely in scientific work. These statistical methods are not worth discussing in this study due to space constraints.

However, the use of complex, long-life sensors requires new solutions in the field of data processing and data analysis and presents unusual problems in this field.

We have the following requirements regarding the data-processing methods of the sensors to be developed:Animal data should be extracted from the primarily measured sensory signal series.Often, the measurement is not taken with the optimal sensor according to the measured characteristic, which makes the evaluation difficult.It has to learn the individual characteristics of each animal.Deviations from the individual characteristics of the animals are detected.Based on the detected deviations, they can start a special measurement program, and send a notification or alert.They can set up a holistic model, based on the differences from the individual’s base level.Experience should be continuously incorporated into the system, so the number of erroneous evaluations may decrease over time.

Several experts agree that the above goals can only be solved using AI algorithms [3,4,62].

### 5.1. Artificial Intelligence Methods Used in Cattle Sensor Papers

Support Vector Machine (SVM) [63,64]

The SVM method separates the labeled training data into two groups according to the type of kernel function, looking for the maximum distance between the two groups to be separated. Classically, it would be used to split a hyperplane in two parts with a hyperline.
(1)$$\begin{array}{c} min\\ f \end{array}{\left\|f\right\|}_{K}^{2}+C\;\overset{l}{\underset{i=1}{\sum }}\left|1-y_{i},f\left(x_{i}\right)\right|,$$
*l* is the number of learning data, *C* is the regularization parameter, and the task is a minimization problem.

K-Nearest Neighbor (KNN) [63]

Similarity is measured with a distance function. The learning database is stored and when an unknown object needs to be classified, we search for the *k* closest points according to the distance function and classify the object into the category that occurs most often among the *k* neighbors (majority voting).
(2)$$\left(\hat{f}x\right)=y_{k,}$$
(3)$$\mathrm{if}\;x_{i}\in N_{k}\left(x\right),$$
where *k* is the number of neighbors, *x_i_* the database, and *y* is the output variable.

Linear discriminant analysis (LDA) [63] is a supervised learning technique, which looks for an orientation that can transform high-dimensional feature vectors to a lower-dimensionality feature space, such that the projected feature vectors of a class on this lower dimensional space remain also distinguishable from the feature vectors of the other classes. LDA is a small-sample-size (SSS) problem in professional terms.

Quadratic discriminant analysis (QDA) [63] is a generalization of LDA, where it cannot be supposed that the covariance matrix in all classes is equal.

Radial base function (RBF) neural network [63] is a feed-forward neural network, which contains three layers (input, hidden, and output layers). It is a function approximator whose learning time is less than traditional MLP-NN.

Multilayer Perceptron (MLP_NN) Neural Network [63]

A fully connected multilayer neural network is called a multilayer perceptron (MLP). It consists of three types of layers, i.e., the input layer, output layer, and hidden layer. MLPs are proven to approximate any continuous function.

Fuzzy solutions and ANFIS models [63]: A fuzzy inference system is based on Zadeh’s theorem and can support complex decisions and function approximations. A fuzzy system works with membership functions and an ANFIS neuro-fuzzy system is able to develop function approximations with the help of the found membership functions.

Deep Convolutional Neural Networks (CNN) [63]

This is a special deep neural network for image processing. It contains frequently at least 20–100 neuron layers with special activation functions.

Based on the literature and our own experiences, it can be stated that in the case of rumen boluses for monitoring purposes, it is worth discussing at least four levels of data processing and implementing them during development.

Level 0 contains the measurement techniques and data preprocessing. The measured data are subjected to various data-cleaning and preprocessing steps, by the end of which raw sensor data are prepared for analysis.

Level 1 covers first-level data processing; extraction of primary-level characteristics; and extraction of biologically interpretable features, during which primary characteristics are obtained from the obtained raw sensor data (rumen temperature, heart rate, type of locomotor activity, body position information, etc.)

Level 2 means complex and integrative health, behavioral, or state assessment at the level of individual animals. It is secondary-level data processing, during which, based on the primary characteristics, by combining them and combining with other information, a result can be obtained that is interpretable for farmers at the level of the individual (estrus detection, time spent ruminating, integral value of locomotor activity, etc.)

Level 3 is a higher, holistic level, representing an integrative big data solution. Big data solutions integrate characteristics of individual animals in a large time scale and at least at the farm level.

### 5.2. Sampling and Preprocessing

The previous chapters highlighted that the use of rumen boluses helps in the continuous monitoring of the animals, thus helping to ensure animal welfare, to reduce the carbon footprint, and to maintain sustainable animal husbandry in general [4]. Unfortunately, developing this bolus with traditional methods is not straightforward. One of the constraints is the limited range of usable sensor modalities described above. This often means that the sensor modality used for the measurement is not optimal. One of the primary purposes of bolus developments is pH measurement; however, lifelong pH monitoring is not yet technically resolved, so it can only be applied for limited periods. Another example is that the characteristics of heart function are easiest to determine with an ECG device, which in turn cannot be used in a lifelong rumen bolus. In practice, with clever signal processing of the data series measured by a three-axis accelerometer, these characteristics (heart rate, heart rate variability, etc. [8]) can be determined. By processing acceleration values, other movement types of the animal (ruminating, general physical activity, etc.) can be separated and quantified. Since all of the mentioned signals are superimposed, determining the different values requires significantly more data-processing steps compared to the data-processing methods of the specialized sensors. This includes preprocessing the measured signals and extracting the valuable signal. Additionally, the orientation of the bolus can be arbitrary inside the rumen, which makes signal processing even more complicated, because physical effects can affect the sensor from different directions.

A detailed description of the data processing was found for the sensor used to test for rumination by 3D acceleration measurements [7]. Data processing is a complex multistep process, with the following steps:Determination of the derivative of the accelerations per axis (jerk).Calculation of the resultant jerk. After this step, the main movement activities can be observed on the graph of the preprocessed values.Additional data-cleaning methods (variance calculation and moving average calculation). This step is required to make further processing more robust.Quantifying the preprocessing data with the appropriate metrics.

These preprocessing steps are used to extract valuable data, and this is followed by special base-level determination and peak detection steps which have already prepared the actual final first-level data processing [7]. An overview of the general data-processing pathway according to acceleration data is given in [65]. It is worth noting that whether we examine rumination movements, accelerations related to heart function (personal communication), or accelerations related to animal activity, the first step in data processing is to determine the derivative function of the acceleration per axis and then the resultant calculation. The preprocessed data obtained during the measurement will usually be the input to the regression calculation or classification. Preprocessing already suggests that the measured values are difficult to interpret due to the large individual variety [5].

The battery life of a rumen bolus does not allow continuous measurement with a high sample with any of the sensors. Communication systems suitable for a rumen bolus are even more constrained in terms of bandwidth and power source (battery life); therefore, it is not possible to transmit large amounts of data continuously. Thus, a well-chosen sampling strategy and statistically based data processing are needed. This approach increases the standard deviation of the features extracted during primary data processing, which makes it difficult to process the data and achieve the desired accuracy classification [1,11,49,65,66,67]. These previous findings suggest that we should use an AI solution at all levels of data processing.

### 5.3. First-Level Data Processing

Regression or classification methods are mainly used in the first-level data processing. In classical research tasks, this is solved by involving a larger number of subjects and statistical tests, but boluses for animal husbandry must handle individual cases. Here, the obvious way to handle plausibility is to use machine-learning algorithms and special normalization methods.

Relatively few studies mention the use of AI algorithms at this level of data processing. Video-based monitoring systems have a much larger literature compared to rumen boluses, and deep-neural-network-based methods are already widely accepted in image processing [1,4,68,69,70,71]. Perhaps unsurprisingly, there are many more AI methodological descriptions here, in the areas of lameness, disease diagnostics, rumination, and activity measurements [1,4,5,65,69]. Another problem with data processing is the significant variability between individuals, which makes classification difficult in data processing. It is often the case that the difference between two healthy individuals is greater than the difference between healthy and sick states or normal and abnormal activity levels of the same individual. This feature has been described in detail in a paper about lameness detection [5]. Lameness can be statistically determined from the locomotor activity of cattle. Location and lying characteristics were used in the experiment, namely median bout duration, step impulses, lying time, number of lying bouts, minimum bout duration, and maximum bout duration. The mean of each measured value is different for lame and non-lame cattle, but the middle two quartiles cannot be separated in any case [5]. The task is complicated by the fact that certain characteristics do not change in the same direction in the case of lameness. The number of steps may decrease because the animal steps less due to the hoof pain, but sometimes, it may take smaller steps, so the number of steps increases. Two important methods were used during preprocessing. The first is that in each case, the sensor learned the normal behavior of the animal during a 14-day learning period, and the actual values of each animal were normalized based on the learned data set during the actual data processing. After that, the measured, preprocessed values of different animals became comparable [5]. The other method is not to examine the measured actual value, but its deviation from the expected value, the latter also on a probabilistic basis.

In classical data processing, the interpretation of the measured characteristics is usually solved with thresholding algorithms, and due to the mentioned difficulties they can only be used with poor efficiency; therefore, it is worth using machine-learning algorithms. In a study on rumination activity detection, the support vector machine (SVM) algorithm was successfully used to determine the rumination activity. The whole task of data processing, adapted to the complex movements measured in the rumen, is quite complex. As the first part of the data processing contained classical data-preprocessing steps and basic statistical methods (moving average, variance), the SVM method was used to evaluate the extracted primary properties, and it was suitable for the determination of rumination [7].

Elements of processing:

1. Peak detector: This is essentially a maximum search method that can detect the 0 value of the derivative function and the change in the sign of the derivative. The peak detector detects all sign changes, even those that belong to a very small peak; thus, a proper selector should be used to select the peaks that are considered appropriate. The peak detector threshold was determined by examining the distribution of the peaks. 2. Intercontraction interval (ICI): The time difference between the peaks marked by peak detectors. 3. Jerk variance baseline (JVB): This is essentially the baseline determination using a rolling median filter in 40 s windows. This time interval was chosen because it corresponds to the length of a rumination period. The two characteristics identified during data preprocessing, ICI and JVB, were inputs to the SVM method. According to the authors, these properties were able to make rumination a linearly separable feature; therefore, SVM was used with a linear kernel. In total, 270s of measured data were required to detect rumination. Based on this, rumination can be detected with an accuracy of 89.2% [7].

Determining rumen temperature is not an easy task. Classical studies have concluded that rumen temperature is not suitable for estimating the body core temperature [16]. This is because the rumen temperature is greatly influenced by the external temperature, the temperature of the water consumed, rumen fermentation, and diseases associated with fiber [10,16,32,33,34,37,40,42,43,45]. We hope that by intelligent baseline estimation, we can obtain rumen temperature that reflects the body temperature. The proposed regression algorithm for baseline determination is a robust matching algorithm [72] specifically recommended for handling one-side deviations [73]. The literature discusses patterns of change in rumen temperature, from which an appropriate sampling frequency can be determined and deviations from baseline can be inferred from feed and water intake, whose number can be estimated as additive information.

Fuzzy solutions can also be used advantageously in the processing of rumen bolus data, although so far, no such publication has come to our attention. It is worth mentioning the fuzzy solutions in other measurements connected to cattle characteristics. One of them allows measuring the weight of moving animals with 1% accuracy. Based on two variables, the measured actual body weight and the animal’s velocity, the model constructs a smoothed body weight curve using a Takagi–Sugeno fuzzy system, the zero frequency Fourier transform of which gives the body weight. Model parameters can be extracted from the ANFIS neuro-fuzzy system and stored in a database [74,75].

### 5.4. Secondary-Level Data Processing

Alsaaod et al. [5] aimed to determine lameness based on locomotor characteristics. The measured data and the preprocessing are described in Section 5.2. During the preprocessing, after the application of a special normalization methodology, the lame and non-lame movements were better separated. However, based on the measured data, linear separability was not assumed. The SVM method was used for the separation, but data were transformed into a linearly separable format using the radial base function (RBF) neural network. The classical elimination technique resulted in an accuracy of only 65% even after parameter optimization, while the SVM method increased the accuracy to 76%.

### 5.5. Holistic Data Processing

It is quite clear to most experts that a holistic approach is needed in precision livestock farming [1,4,13,62,67,71,76,77]. This can be achieved by integrating and analyzing data from different data sources together. The data-driven perspective has been the focus of several publications. A review has also been conducted in this area, summarizing the data integration methods and algorithms used, and detailing the goals and potential of this approach [63]. It is now clear that, as well as at the individual-animal-monitoring level or at the farm level, data integration includes data from monitoring IoT devices and the use of other, more traditional data sources. Rumen boluses have not yet been investigated in studies in terms of their role and significance in data integration. It has become clear that only AI algorithms can achieve good results in this area. On the second and third level of data processing the suggested methods, there are the computationally more intensive deep neural network algorithms. Now, these methods are more and more widespread in any application (including from self-driving cars to smart buildings) [78]. In this area, the most frequent applications are the video-based data analysis methods. However, the video cameras are not able to be applied comfortably in stables, and their usage is expected to be limited in the future, but their data-processing algorithms are pioneering in precision agriculture. It is expected that these type of algorithms (e.g., CNN) with many modern techniques, such as network pruning, weight quantization, and transfer learning, will become important in data processing [79,80]. Other proposed algorithms are fuzzy, neuro-fuzzy systems, and dimension reduction methods used in big data systems, or deep neural network-based solutions [1,4,49,57,67]. It is important to mention the ongoing need and the solutions that are coming to the fore today for the interpretability of the AI models. This has been a classically large problem for AI systems; however, with the proliferation of explainable AI, the models can be interpreted, helping the proposed interventions obtain more user acceptance [81,82].

### 5.6. Systematic Review on Cattle Bolus Artificial Intelligence Methods

In a systematic literature search, we searched the Google Scholar and ScienceDirect databases for the keyword combination “cattle” and “bolus” and “artificial intelligence” and “cattle” and “bolus” and “Machine learning”. As a result of the search, we obtained Table 2. The acronym AI could not be used in the search because it is already included in animal husbandry jargon.

It is clear that the number of hits increases over time, indicating the importance of this area.

The article by Bradhurst et al. [68] used AI solutions at the third level of analysis in the study of epidemic spread. In the article, in fact, the bovine rumen bolus occurs only at the mentioned level [83].

The article by Campos et al. [69] used machine-learning algorithms in the second level of analysis. Authors investigated the nutritional characteristics of goats using an sEMG sensor. The bovine rumen bolus appears in the paper as a mention, but the authors recognized that AI algorithms should be used for classification in sensor-based analyses. The paper includes a comparison of K nearest neighbor (KNN), linear discriminant analysis (LDA), quadratic discriminant analysis (QDA), support vector machine (SVM), multilayer perceptron neural network (MLP-NN), and radial base function (RBF) methods. It notes that the highest classification accuracy in the given analysis could be achieved with the MLP-NN model [6].

Experimental rumen bolus temperature measurement A: M. Lees (2019), mostly using classical statistical methods for analysis, only mentions AI algorithms without the extent of their use [16]. Another paper specifically discusses IoT sensors used on animal farms and applicable AI solutions [3]. One article deals with a rumen bolus that can be used to detect rumination, using SVM classification in the analysis [7]. Rumination was detected via neck-mounted accelerometers and a RumiWatch halter [84].

Analysis of bovine behavior was conducted by Fuentes et al. (2020) using data from a video system and using a deep neural network. The methodology used is an alternative to the rumen bolus technique, and the referenced publication can be used to recognize and subdivide 15 different types of hierarchical behaviors [70]. Jiang’s study also includes an analysis of video data that examines the identification of bovine lameness. The study uses a convolutional neural network with an accuracy of over 90%.

Knight’s analysis (2020) is a review [1] of sensor techniques; this study also considers rumen boluses to be a forward-looking approach and includes an analysis of the significance of the artificial intelligence solutions that can be used, without detailing the specific uses of each solution. Bilali’s analysis deals with the subject in a similar way [4].

The year 2021 stood out in integrative data analysis. Ramadhan et al. [13] highlighted the importance of the continuous e-monitoring of livestock, mentioning rumen bolus applications and artificial intelligence methods in modern animal husbandry among modern technologies. Cabrera similarly analyzes the possibilities of data analysis and data integration in the precision dairy sector in a systematic literature search. The paper specifically addresses the machine learning algorithms used in holistic data analysis [67].

## 6. Conclusions

Rumen bolus sensors appear to be a promising technology in the dairy cattle sector. Today, modern animal husbandry increasingly requires continuous, practically lifelong monitoring of animals which promises significant economic and animal welfare benefits. This can be ensured by the rumen bolus sensor due to its stable location and the adequate lifespan of the batteries.

In this article, we reviewed the results achieved so far in this field, starting with the physical and hardware solutions of the devices, continuing with the possible ways of collecting information, applicable sensory modalities, and the results of information transmission and processing. Based on the reviewed literature, it can be stated that some AI solutions are already used in current devices, but this area still has many possibilities. Straightforward results have been reported in the preprocessing and individual-level evaluation of sensor data using the algorithms of machine learning. The use of SVM, random forest, KNN and multilayer perceptron as well as RBF neural networks has yielded significant results in the literature. Deep-learning methods are advantageous in the processing of image and video data; however, continuous monitoring with a camera is a technically difficult solution. Nevertheless, the statistical methods used also serve as a lesson for sensor monitoring. In the future, the range of solutions used will certainly expand. It will be worthwhile to test the intelligent classification and regression techniques, the ‘K nearest neighbor’ method, the ‘random forest’, and the Bayesian classification methodology during temperature measurement. Fuzzy methods are also promising in data processing. Holistic data analysis is typically a task based on big data and requires complex analysis, where deep-learning algorithms can play a significant role. The weak point of AI, and in particular neural-network-based solutions, is the interpretability of the results, which is a fundamental issue of trust for farmers. In our opinion, explainable AI solutions might help interact with data analytics and can make this method acceptable to farmers as well.

## Figures and Tables

**Figure 1 sensors-22-06812-f001:**
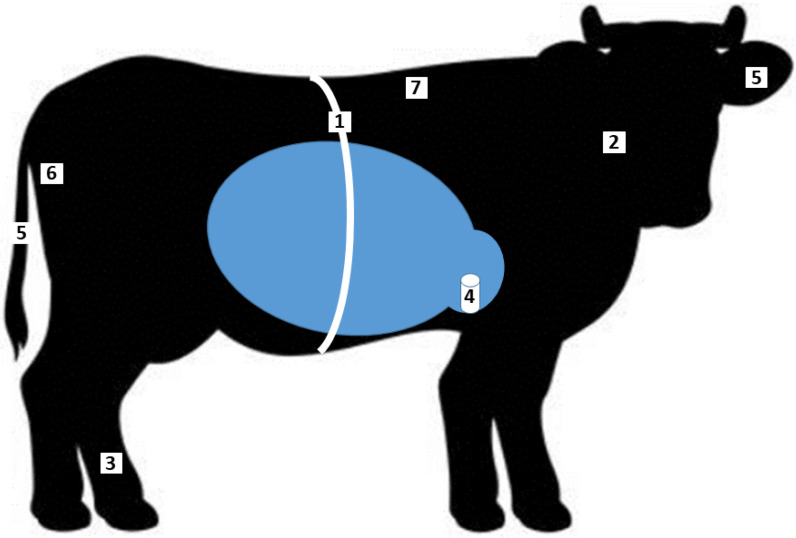
Dairy cattle “at cow” sensors according to Table 1. 1. Body surface ECG equipment, 2. Neck rumination sensor, 3. Leg activity sensor, 4. Rumen sensors, 5. Tail, ear estrus detector, 6. Vaginal partition detector, 7. Subcutaneous sensor.

**Table 1 sensors-22-06812-t001:** Sensor groups according to the placement location and methodology, usage length, aim of data processing.

	Sensor Type by Placement	Placement without Hurt	Usage Length	Aim of Data Processing
1	Body surface ECG equipment	+	1 to 2 days	processed measurement result
2	Neck rumination sensor	+	few weeks–3 years	processed measurement result
3	Leg activity sensor	+	1–3 weeks	processed measurement result
4	Rumen sensors, originally pH measurements	+	3 months–5 years	processed measurement result, complex expert opinion/alert
5	Tail, ear estrus detector	+	few days	processed measurement result complex expert opinion/alert
6	Vaginal partition detector	+	up to 1 week	processed measurement result complex expert opinion/alert
7	Subcutaneous sensor	-	5 years	processed measurement result, complex expert opinion/alert

**Table 2 sensors-22-06812-t002:** Search and relevant result of keywords: “cattle” and “bolus” and “Artificial Intelligence” and “cattle” and “bolus” and “Machine learning”.

Year	Google Scholar	Science Direct
2016	12	1 [83]
2017	18	0
2018	34	1 [6]
2019	42	3 [3,7,16]
2020	60	2 [1,70]
2021–2022	132	6 [13,62,65,67,76,77]

## Data Availability

Not applicable.

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
