# Peer review of "Dairy Cattle Rumen Bolus Developments with Special Regard to the Applicable Artificial Intelligence (AI) Methods"

_sensors, 2022, doi:10.3390/s22186812_

Round 1

Reviewer 1 Report

The work is interesting, the topic is important for the cattle breeding market, and the information is relevant and well organized. I suggest extending the information on "Data transfer solutions", currently there are many systems more adjusted to IoT than WiFi and BLE, such as LoRa, sigfox, among others. Additionally, I believe that commercial devices should be mentioned and referenced so that the work is also useful for people who only need to implement and not from the point of view of research.

Author Response

The authors are grateful for the efforts of Reviewers in the evaluation of the manuscript. We appreciate your time spent with the review. We feel that the Reviewers’ comments and recommendations were reasonable, and we tried to take them into account as far as possible while improving the manuscript. The activities of both Reviewers contributed significantly to the improvement of the quality of our paper.

AU: According to the Editor’s recommendation, we rephrased lines 121-144, 160-164 and 299-303.

Referee 1's comments to Authors:

The work is interesting, the topic is important for the cattle breeding market, and the information is relevant and well organized. I suggest extending the information on "Data transfer solutions", currently there are many systems more adjusted to IoT than WiFi and BLE, such as LoRa, sigfox, among others. Additionally, I believe that commercial devices should be mentioned and referenced so that the work is also useful for people who only need to implement and not from the point of view of research.

AU: Data transfer solution chapter was extended with the suggested technologies and solutions.

Reviewer 2 Report

1. Deep learning neural network has shown much more powerful capability than KNN, SVM, LDA methods. Therefore, this paper should cite more papers in using deep learning neural networks for monitoring purpose, such as

[1] Q. Huang, K. Hao, "Development of CNN-based visual recognition air conditioner for smart buildings", Journal of Information Technology in Construction, vol. 25, pp. 361-373, 2020.

2. The use of deep learning neural networks consumes million of trainable parameters. To deal with this challenge, some techniques are proposed, such as network pruning, weight quantization, transfer learning. Do you use these techniques to reduce the memory usage and complexity of your AI method?

[1] Z. Tang et al., "Automatic Sparse Connectivity Learning for Neural Networks," in IEEE Transactions on Neural Networks and Learning Systems, doi: 10.1109/TNNLS.2022.3141665.

[2] J. Zheng, C. Lu, C. Hao, D. Chen and D. Guo, "Improving the Generalization Ability of Deep Neural Networks for Cross-Domain Visual Recognition," in IEEE Transactions on Cognitive and Developmental Systems, vol. 13, no. 3, pp. 607-620, Sept. 2021, doi: 10.1109/TCDS.2020.2965166.

Author Response

The authors are grateful for the efforts of Reviewers in the evaluation of the manuscript. We appreciate your time spent with the review. We feel that the Reviewers’ comments and recommendations were reasonable, and we tried to take them into account as far as possible while improving the manuscript. The activities of both Reviewers contributed significantly to the improvement of the quality of our paper.

AU: According to the Editor’s recommendation, we rephrased lines 121-144, 160-164 and 299-303.

Referee 2’s comments to the Authors:

  1. Deep learning neural network has shown much more powerful capability than KNN, SVM, LDA methods. Therefore, this paper should cite more papers in using deep learning neural networks for monitoring purpose, such as

[1] Q. Huang, K. Hao, "Development of CNN-based visual recognition air conditioner for smart buildings", Journal of Information Technology in Construction, vol. 25, pp. 361-373, 2020.

  1. The use of deep learning neural networks consumes millions of trainable parameters. To deal with this challenge, some techniques are proposed, such as network pruning, weight quantization, transfer learning. Do you use these techniques to reduce the memory usage and complexity of your AI method?

AU: The "5.5 Holistic data processing" chapter was extended with the suggested content and the suggested literature was also referenced.

[1] Z. Tang et al., "Automatic Sparse Connectivity Learning for Neural Networks," in IEEE Transactions on Neural Networks and Learning Systems, doi: 10.1109/TNNLS.2022.3141665.

[2] J. Zheng, C. Lu, C. Hao, D. Chen and D. Guo, "Improving the Generalization Ability of Deep Neural Networks for Cross-Domain Visual Recognition," in IEEE Transactions on Cognitive and Developmental Systems, vol. 13, no. 3, pp. 607-620, Sept. 2021, doi: 10.1109/TCDS.2020.2965166.

Round 2

Reviewer 2 Report

The quality of this work has been improved a lot.